# Differences in pathways contributing to thyroid hormone effects on postnatal cartilage calcification versus secondary ossification center development

**Gustavo A Gomez[1], Patrick Aghajanian[2], Sheila Pourteymoor[1], Destiney Larkin[1], Subburaman Mohan[1,3,4,5]***

[1]Musculoskeletal Disease Centre, Jerry L. Pettis VA Medical Center, Loma Linda, United States; [2]Fulgent Genetics, El Monte, United States; [3]Departments of Medicine, Loma Linda University, Loma Linda, United States; [4]Departments of Biochemistry, Loma Linda University, Loma Linda, United States; [5]Departments of Physiology, Loma Linda University, Loma Linda, United States

**Abstract** The proximal and distal femur epiphyses of mice are both weight-bearing structures derived from chondrocytes but differ in development. Mineralization at the distal epiphysis occurs in an osteoblast-rich secondary ossification center (SOC), while the chondrocytes of the proximal femur head (FH), in particular, are directly mineralized. Thyroid hormone (TH) plays important roles in distal knee SOC formation, but whether TH also affects proximal FH development remains unexplored. Here, we found that TH controls chondrocyte maturation and mineralization at the FH in vivo through studies in *thyroid stimulating hormone receptor* (*Tshr*[-/-]) hypothyroid mice by X-ray, histology, transcriptional profiling, and immunofluorescence staining. Both in vivo and in vitro studies conducted in ATDC5 chondrocyte progenitors concur that TH regulates expression of genes that modulate mineralization (*Ibsp*, *Bglap2*, *Dmp1*, *Spp1*, and *Alpl*). Our work also delineates differences in prominent transcription factor regulation of genes involved in the different mechanisms leading to proximal FH cartilage calcification and endochondral ossification at the distal femur. The information on the molecular pathways contributing to postnatal cartilage calcification can provide insights on therapeutic strategies to treat pathological calcification that occurs in soft tissues such as aorta, kidney, and articular cartilage.

*For correspondence:
Subburaman.Mohan@va.gov

## Editor's evaluation

This paper describes the differential effects of thyroid hormone on chondrocyte maturation and mineralization at the femoral head in mice compared with the distal femur. The comprehensive set of studies were carried out in *Tshr*[-/-] hypothyroid mice through radiologic and histologic methods as well as transcriptional profiling. The conclusions are of value in understanding bone growth defects during hypothyroid states.

## Introduction

Bones make up the infrastructure of the body and are formed through a process known as ossification. Most bones are formed by endochondral ossification where condensed mesenchymal stem cells proliferate and differentiate into chondrocytes. Ossification first occurs in the mid-shaft of the bone, which forms the primary center of ossification (POC) and expands toward the ends of the cartilage

matrix. In mice, the POC has been established to occur during embryonic development day (E) E14.5–15.5 while the secondary ossification center (SOC) forms at approximately postnatal day (P) P5–7 at the epiphyseal ends (*Mackie et al., 2011*). Each stage of skeletal development from chondrocytes can be characterized by the expression level of specific genes. Early chondrocytes which are characterized by expression of collagen, type 2, alpha 1 (COL2A1), and aggrecan (ACAN) undergo proliferation, withdraw from the cell cycle, and differentiate into pre-hypertrophic chondrocytes expressing Indian hedgehog (IHH) and Sp7 transcription factor (SP7). Hypertrophic chondrocytes express high levels of a cartilage matrix consisting of matrix metalloproteinase 13 (MMP13) and collagen, type 10, alpha 1 (COL10A1), creating a template for bone formation. Expression of vascular endothelial growth factor A (VEGFA) subsequently leads to invasion by capillaries, which allows the influx of osteoblasts, osteoclasts, and bone marrow cells to replace the cartilage matrix with mineralized bone (*Li and Dong, 2016*; *Mackie et al., 2011*; *Takarada et al., 2016*). The growth plate is responsible for longitudinal skeletal growth and skeletal maturity is reached when the POC and SOC meet (*Mackie et al., 2011*).

Thyroid hormone (TH) is an important regulator of skeletal growth and development. Optimal levels of TH peak simultaneously with the initiation of SOC formation and are essential for its development (*Aghajanian et al., 2017*; *Kim and Mohan, 2013*; *Xing et al., 2014*). Dysregulation in the amount of TH during skeletal development can lead to growth arrest, and delayed bone formation in both humans and mice (*Bassett et al., 2008*; *Gogakos et al., 2010*; *Kim and Mohan, 2013*). Previous work from our group has substantiated the importance of TH in bone formation. We found that TH regulates the development of the SOC through IHH signaling and SP7 activity, and that TH is a major regulator of a number of key bone growth factors, including insulin-like growth factor-I (*Mohan et al., 2003*; *Wang et al., 2010*; *Wang et al., 2006*). We also established that TH-deficient $Tshr^{-/-}$ mice have severely compromised development of the epiphysis in both the femur and tibia at the knee joint, which is completely rescuable by TH treatment for 10 days when serum levels of TH rise in wild type mice (*Xing et al., 2014*). Additionally, we found that TH regulates SOC formation at the epiphysis of the distal femur and proximal tibia, by a TH-induced chondrocyte-to-osteoblast transdifferentiation mechanism (*Aghajanian et al., 2017*).

Previous studies have revealed that bone development culminates at a later timepoint in the proximal femur relative to the distal femur, suggesting a difference in developmental mechanisms (*Patton and Kaufman, 1995*). A recent study in mice by *Cole et al., 2013*, found that the proximal and distal femur had different developmental patterns in terms of timing, vascular development, and ossification. While the cartilage template at the distal epiphysis was replaced by bone matrix via a chondrocyte-to-osteoblast transdifferentiation-mediated endochondral ossification process, at the proximal femoral epiphysis cartilage was directly mineralized without the involvement of an SOC (*Cole et al., 2013*). In this work, we examined whether TH is also involved in regulating cartilage mineralization at the proximal femur epiphysis and the mechanisms for the differential effects of TH-mediated endochondral ossification at the epiphyseal structures of the knee versus direct cartilage mineralization at chondrocytes of the femur head (FH). Our findings demonstrate for the first time the mechanisms that contribute to differential development of the proximal versus distal femur, providing novel information on the physiological and conceivably, pathological means of soft tissue mineralization.

## Results

### TH is necessary for mineralization of the proximal FH

In order to determine whether the cartilage mineralization that occurs at the proximal femur of mice (*Cole et al., 2013*) is dependent on TH signaling, we evaluated the hip joints of hypothyroid $Tshr^{-/-}$ and euthyroid $Tshr^{+/-}$ on day 21. X-ray imaging of genotyped mice revealed that compared with the tight apposition of the FH with the acetabulum of the pelvic bone in $Tshr^{+/-}$, a distinct gap was evident between the FH and acetabulum in $Tshr^{-/-}$ mice (*Figure 1A*). Given that injection of TH on days 5–14 restores the SOC defect at the distal femur (*Xing et al., 2014*), we tested whether the defect observed at the proximal femur is also directly dependent on TH. While all three of the proximal femur structures (greater trochanter, FH, and lesser trochanter) were underdeveloped in $Tshr^{-/-}$ injected with vehicle compared to $Tshr^{+/-}$ mice (*Figure 1B*), restoration of all three structures in $Tshr^{-/-}$ injected with T3/T4 indicates TH signaling also regulates development and mineralization of the proximal femur epiphysis (*Figure 1C*).

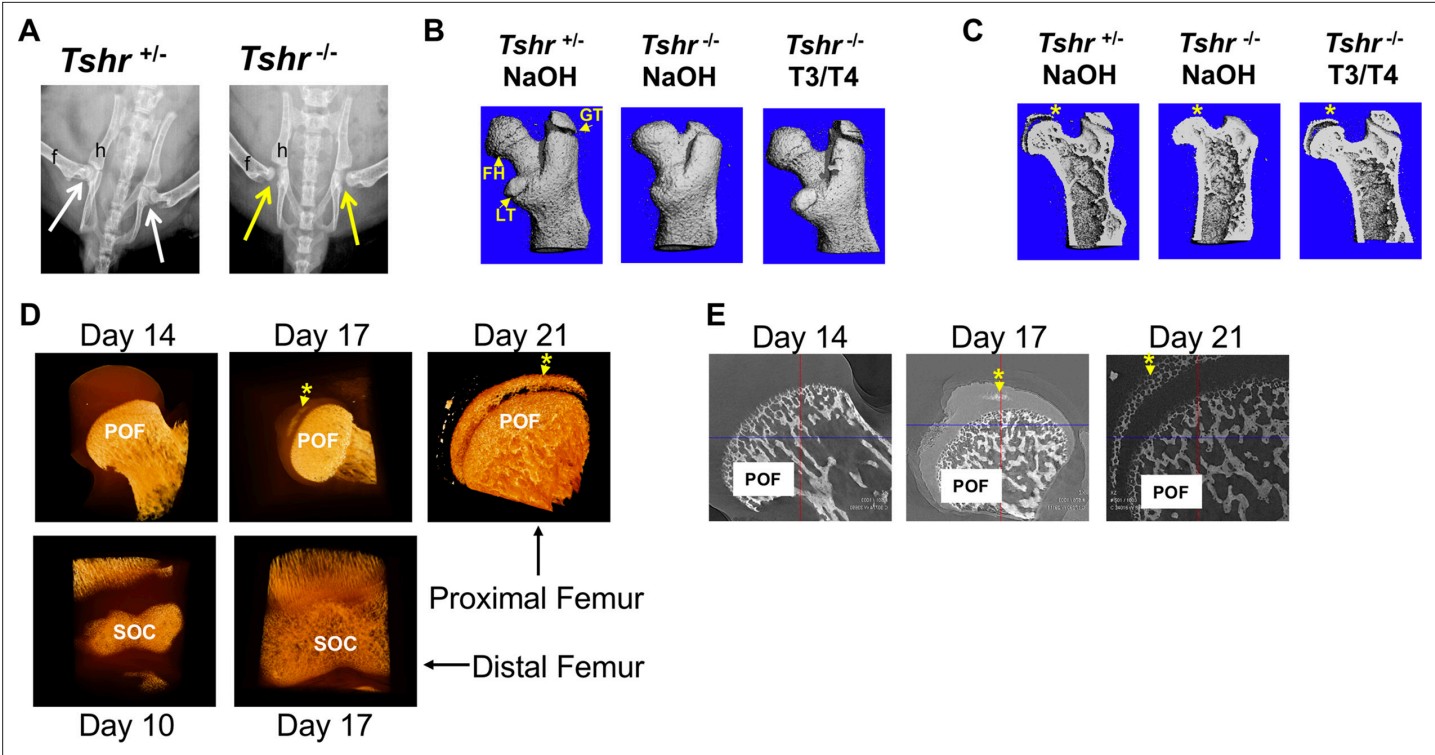

**Figure 1.** *Tshr-/-* proximal femur phenotype and onset of mineralization. (Panel A) Ventral X-ray view of postnatal (P) 21-day-old mice. Arrows point to joint between hip (h) and femur (f). (Panel B) Lateral view of three-dimensional (3D) μCT scans from P21 proximal femurs. FH = femur head, GT = greater trochanter; LT = lesser trochanter. (Panel C) Lateral μCT hemi-section view of proximal femurs. Asterisk adjacent to region mineralized in FH. (Panel D) Lateral views of 3D nano-computed tomography (nano-CT) images of proximal FH (top row) and distal femur epiphysis (bottom row). Mineralized tissue is opaque ivory/gold colored. (Panel E) Two-dimensional sections of nano-CT. Mineralized tissue is white/bright gray. POF: primary ossification front, SOC: secondary ossification center. Asterisk and arrow point to mineralization at FH. Images shown are representative of three to five mice per group.

Although mineralization of both distal femur and proximal tibia epiphysis initiates by the end of the first week after birth in mice (*Aghajanian et al., 2017*; *Mackie et al., 2011*), the time at which mineralization initiates in the proximal femur epiphysis remains poorly defined. Consequently, we evaluated the earliest appearance of mineralization in proximal FH cartilage in *Tshr+/-* mice by high-resolution nano-computed tomography (nano-CT). Our data revealed a 10-day delay in the initiation of mineralization at the FH compared to the distal femoral epiphysis (*Figure 1D and E*).

## FH cartilage mineralization is delayed in hypothyroid mice

To further characterize the tissue mineralized at the FH, longitudinal sections of proximal femurs of euthyroid *Tshr+/-* and hypothyroid *Tshr-/-* mice were compared at P10 and P21 by histology staining for cartilage (Safranin O and toluidine blue), bone (Von Kossa), and mineral (Alizarin red). Relative to *Tshr+/-* controls, the cartilage area was greater at the proximal FH in *Tshr-/-* mice (*Figure 2A and B*). In the proximal FH, Alizarin red and Von Kossa mineral staining were only detected on day 21 of *Tshr+/-* controls (*Figure 2C and D*), which corroborates nano-CT results. The proximal FH of *Tshr-/-* is eventually mineralized in more mature mice (*Figure 2—figure supplement 1*). Conversely, extensive mineral staining was seen at the distal femur SOC in euthyroid mice at both timepoints, but to a reduced extent in the distal femur SOC of *Tshr-/-* mice on day 21 (*Figure 2C and D*). Interestingly, positive staining for tartrate resistant acid phosphatase (TRAP, also known as ACP5) was found in the actively mineralizing region of distal femur SOC (*Figure 2E*), which was also dependent on TH status. However, TRAP staining was not detected in the FH of either genotype, even on day 21. These data suggest that while TH seems to play an important role in the mineralization of proximal FH as in the case of the distal femur, there are important differences in when mineralization occurs, and the type of tissue being mineralized at both ends.

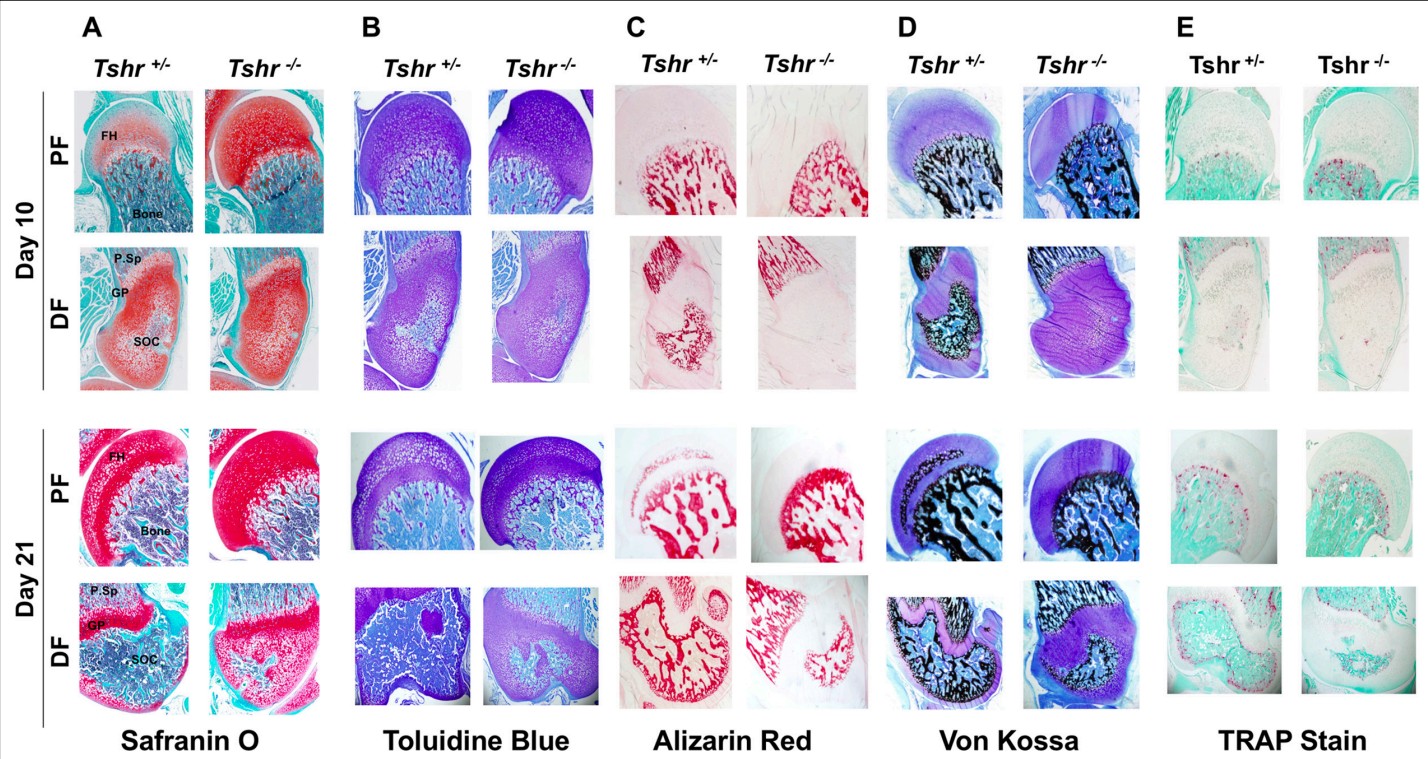

**Figure 2.** Histological analysis of femur head (FH) in euthyroid *Tshr*[+/-] and hypothyroid *Tshr*[-/-] mice. Safranin O stained sections showing cartilage in red (Panel A), toluidine blue stained sections showing cartilage in violet (Panel B), Alizarin red stained sections showing mineral in red (Panel C), Von Kossa stained sections showing mineral in black (Panel D), and TRAP/ACP5 stained sections showing TRAP activity in red (Panel E) at the proximal femur, PF, and distal femur, DF, from *Tshr*[+/-] and *Tshr*[-/-] mice. P.Sp = primary spongiosa; GP = growth plate; SOC = secondary ossification center.

The online version of this article includes the following figure supplement(s) for figure 2:

**Figure supplement 1.** Delayed mineralization in *Tshr*[-/-] femur head (FH).

## Transcription profiles reveal delay in chondrocyte maturation in FH of *Tshr*[-/-] mice

Given that TH elicits distinct responses at proximal FH and distal femur epiphysis, which both develop from chondrocytes, we compared transcriptional changes of genes involved in chondrocyte/osteoblast maturation and ECM remodeling at both sites on days 10 and 21 between *Tshr*[+/-] and *Tshr*[-/-] mice that were treated with or without TH by reverse transcriptase quantitative polymerase chain reaction (RT-qPCR). A delay in maturation at both ends in day 10 hypothyroid femurs was evident by reduced mRNA levels of genes expressed in mature chondrocytes *Ihh*, *Tnfsf11*, *Sp7*, *Col10a1*, *Alpl*, and *Mmp13* (*Figure 3A–D*). Moreover, markers of immature chondrocytes *Shh* and *Sox9* were increased, but only significantly at the FH (*Figure 3A and B*), indicating a more substantial delay in maturation of FH chondrocytes on day 10. TH treatment increased *Col10a1* expression in the distal femur but not proximal FH at day 10, further supporting a delay in chondrocyte maturation at the FH.

Furthermore, while expression levels of genes involved in mineralization, *Ibsp*, *Spp1* and *Acp5*, were reduced, *Mgp* expression was elevated at the distal femur of day 10 *Tshr*[-/-] mice, but none of these were affected in the FH (*Figure 3C and D*). These data are consistent with the histology data demonstrating active mineralization at distal but not proximal FH at this timepoint. Since others, and we, have shown a key role for hypoxia signaling in chondrocyte maturation (*Cheng et al., 2016*; *Cheng et al., 2017*; *Schipani et al., 2001*; *Yellowley and Genetos, 2019*), we measured expression of hypoxia signaling genes and found that in day 10 *Tshr*[-/-] femurs the expression levels of *Hif1a* were unchanged at both the FH and distal femur, while *Epas1* and *Vegfa* were reduced at the FH and not restored by TH treatment (*Figure 3A and B*). *Tgfa* expression was negatively regulated by TH at the FH but not at the distal femur (*Figure 3A*), thus suggesting region-specific effects of TH on the femur.

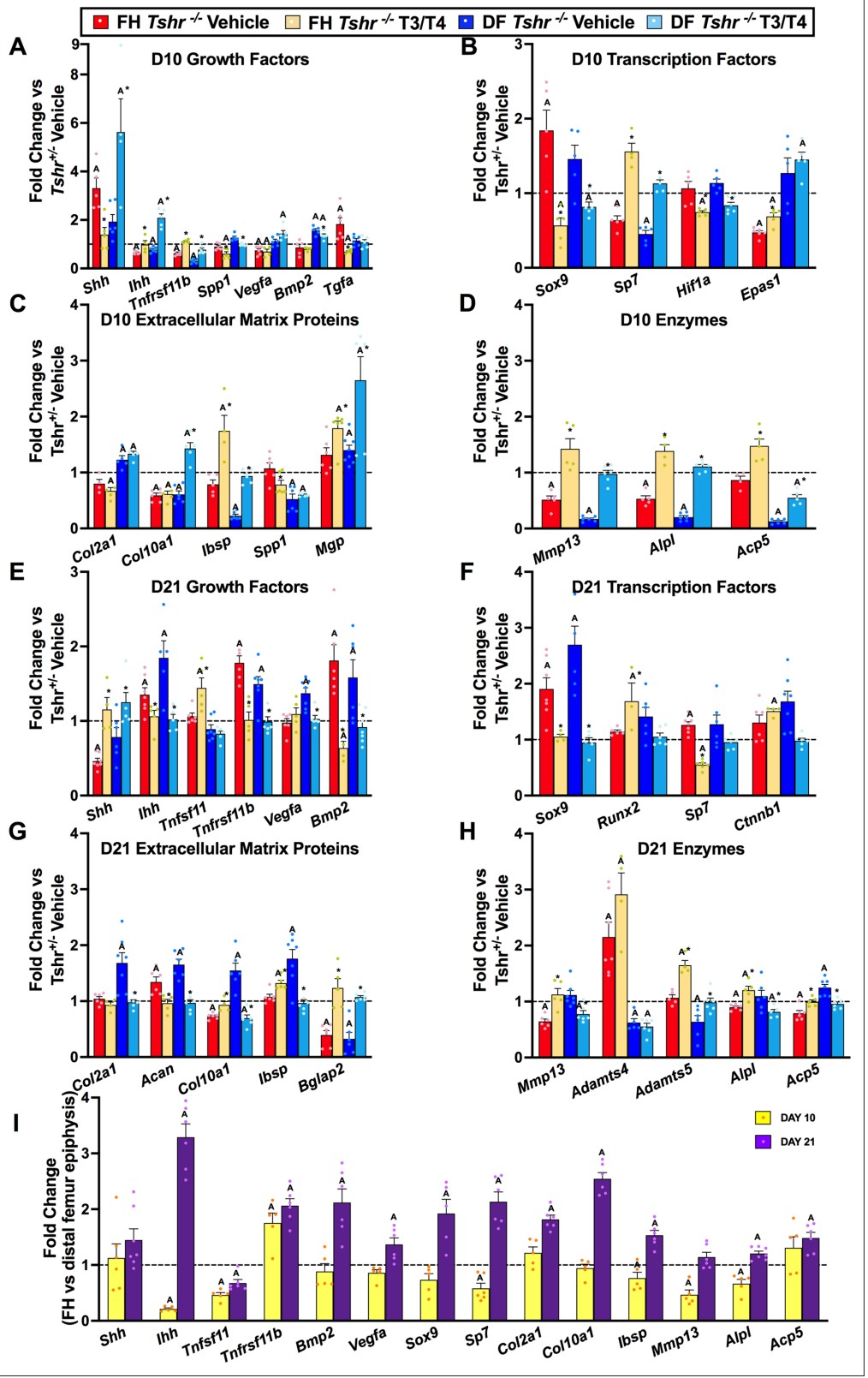

**Figure 3.** Thyroid hormone (TH)-dependent transcriptional differences at proximal and distal femur (DF). (Panels A–G) Gene expression changes from femur head (FH) or DF between *Tshr*[-/-] injected with vehicle, or T3/T4, plotted as fold-change relative to anatomical and stage matched region in euthyroid *Tshr*[+/-] with vehicle (dashed line). Day 10 samples (Panels A–D) were treated on days 5–9, while day 21 samples (Panels E–G) were treated on days

*Figure 3 continued on next page*

*Figure 3 continued*

5–14. (Panel I) Fold-change of mRNA expression at FH versus DF epiphysis in *Tshr*[+/-] days 10 and 21. Statistics analyzed by t-test where A = p < 0.05, * = p < 0.05 between T3/T4 treatment and vehicle at FH or DF. (n = 6) Sonic hedgehog (*Shh*), Indian hedgehog (*Ihh*), tumor necrosis factor receptor superfamily member 11b – synonym rank ligand (*Tnfsf11b/Rankl*), tumor necrosis factor receptor superfamily member 11b – synonym osteoprotegerin (*Tnfrsf11b/opg*), vascular endothelial growth factor a (*Vegfa*), bone morphogenetic protein 2 (*Bmp2*), transforming growth factor alpha (*Tgfa*), SRY-box transcription factor 9 (*Sox9*), runt-related transcription factor 2 (*Runx2*), Sp7 transcription factor – synonym osterix (*Sp7/Osx*), hypoxia inducible factor 1alpha (*Hif1a*), endothelial PAS domain protein 1 – synonym hypoxia inducible factor 2alpha (*Epas1/Hif2a*), collagen type 2 alpha 1 (*Col2a1*), collagen type 10 alpha 1 (*Col10a1*), integrin binding sialoprotein – synonym bone sialoprotein (*Ibsp/bsp*), secreted phosphoprotein 1 – synonym osteopontin (*Spp1/Opn*), bone gamma-carboxyglutamate protein 2 – synonym osteocalcin (*Bglap2/Ocn*), matrix gla protein (*Mgp*), matrix metallopeptidase 13 (*Mmp13*), alkaline phosphatase liver/bone/kidney (*Alpl*), acid phosphatase five tartrate resistant – synonym tartrate resistant acid phosphatase (*Acp5/Trap*), beta catenin (*Ctnnb1*), aggrecan (*Acan*), ADAM metallopeptidase with thrombospondin type 1 motif 4 (*Adamts4*), ADAM metallopeptidase with thrombospondin type 1 motif 5 (*Adamts5*).

The online version of this article includes the following source data for figure 3:

**Source data 1.** Source data for *Figure 3A–D*.

**Source data 2.** Source data for *Figure 3E–H*.

**Source data 3.** Source data for *Figure 3I*.

On day 21, femurs of hypothyroid mice continued to display profiles suggesting a delay in maturation. Expression levels of immature chondrocyte markers, *Sox9* and *Acan*, were increased at both ends, yet reduced by TH treatment (*Figure 3F and G*). While *Col2a1* mRNA levels were unchanged, *Col10a1* mRNA levels were reduced in the FH of *Tshr*[-/-] mice (*Figure 3G*). Increased mRNA levels of *Ihh* in both proximal and distal femurs of *Tshr*[-/-] mice were also restored to control levels by TH treatment, thus suggesting a role for *Ihh* in chondrocyte maturation. *Alpl* and *Acp5* transcripts were reduced at the proximal FH but not distal femur of *Tshr*[-/-] mice at day 21 and rescued by TH (*Figure 3H*). TH treatment produced opposite effects on *Ibsp* expression at the two femoral sites in *Tshr*[-/-] mice on day 21. By contrast, the reduced *Bglap2* mRNA levels at the proximal FH and distal femur of *Tshr*[-/-] mice at day 21 were rescued by TH treatment (*Figure 3G*). Interestingly, expression levels of growth factors *Bmp2* and *Tnfrsf11b* were elevated at both ends of the femur in *Tshr*[-/-] mice on day 21 and rescued by TH treatment (*Figure 3E*). Although expression of *Runx2* and *Ctnnb1* was not affected at either FH or distal femur (*Figure 3F*), TH treatment increased their expression at the FH only. Thus, while TH is required for continuous maturation of chondrocytes and affects common pathways, it can also differentially regulate distinct genes at the proximal FH and distal femur epiphysis.

To further characterize temporal changes of chondrocyte maturation at FH and distal femur, we compared changes in expression levels of genes in the FH relative to stage matched distal femurs on days 10 and 21 in *Tshr*[+/-] mice. The reduced expression of *Ihh*, *Tnfsf11*, *Sp7*, *Mmp13*, *Ibsp,* and *Alpl* (*Figure 3I*) at day 10 was consistent with the delayed maturation of chondrocytes at the proximal FH. However, at day 21, many of these genes were expressed at higher levels in the FH, a finding consistent with active mineralization occurring at this time. Therefore, transcriptional profile data reveals that chondrocytes are at a relatively more immature state in the FH of day 10 compared to day 21, and there was a catch-up in maturation as noted by increased expression of genes associated with chondrocyte maturation and mineralization in the FH on day 21.

## Spatiotemporal profile of protein expression in FH chondrocyte development

To further characterize the molecular mechanisms that contribute to the development of the FH, we performed a time-course spatiotemporal analysis of proteins that report different stages of chondrocyte and osteoblast maturation by immunohistology on *Tshr*[+/-] days 10, 17, and 21, and in *Tshr*[-/-] on day 17, when FH mineralization commences. Antibody fidelity was determined by spatial domains of expression in the proximal tibia of day 10 *Tshr*[-/-], where active bone mineralization is underway, and occurs identically as in the distal femur.

We previously showed that expression of collagen proteins follows a linear progression of appearance whereby COL2A1 secretion by immature chondrocytes is followed by COL10A1 in

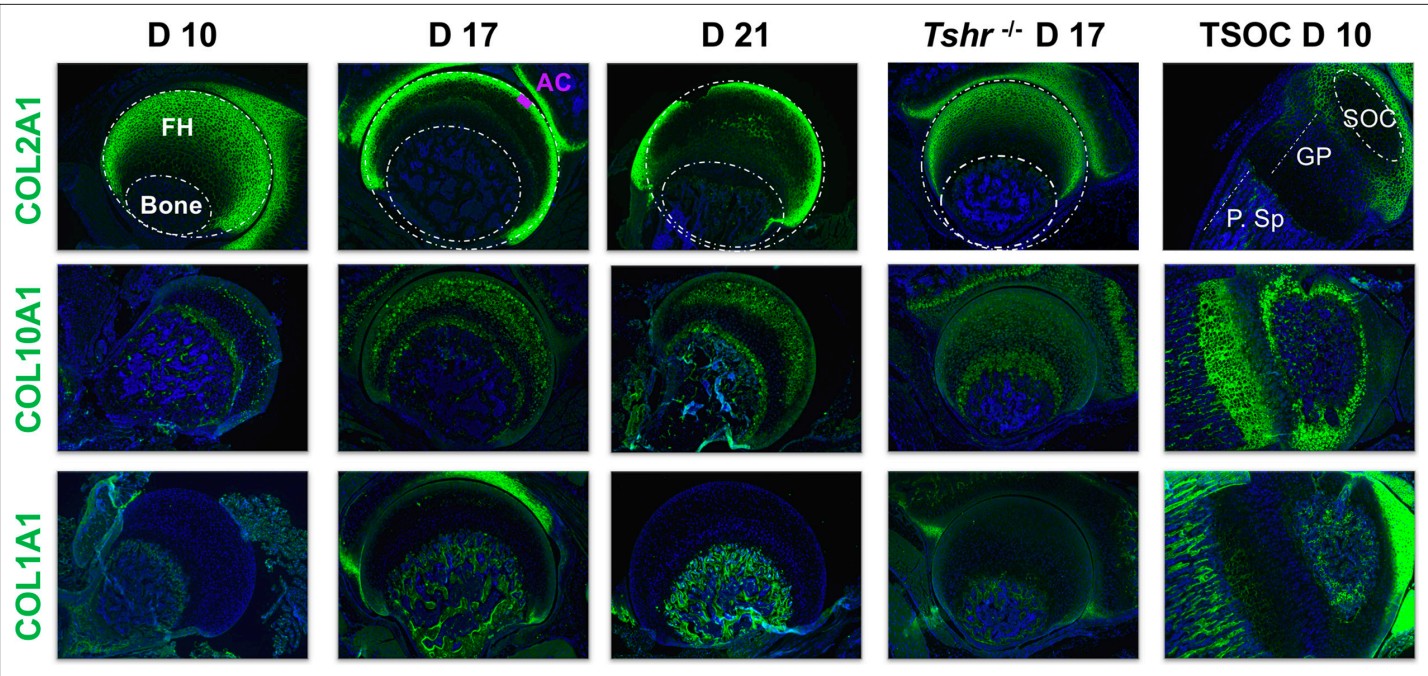

**Figure 4.** Immunohistochemical characterization of femur head (FH) development. Longitudinal sections of *Tshr⁺/⁻* were probed for protein expression in FH on days 10, 17, 21, proximal tibia epiphysis on day 10, and FH of *Tshr⁻/⁻* on day 17 by immunofluorescence for collagens: COL2A1, COL10A1, COL1A1 (all green). Immunofluorescent images counterstained with DAPI (blue). Abbreviations are references for all immunohistology figures (***Figures 4–9***); FH, femur head; AC, articular cartilage; SOC, secondary ossification center; GP, growth plate; P.Sp, primary spongiosa.

pre-hypertrophic and hypertrophic chondrocytes, and COL1A1 in mineralizing osteoblasts during the second week of postnatal life at the distal femur and proximal tibia (***Aghajanian et al., 2017***; ***Xing et al., 2014***). In the proximal FH, however, there was a delay in the expression of markers of chondrocyte differentiation as noted by the presence of COL2A1 in chondrocytes of day 10 FH, and subsequent replacement of COL2A1 with COL10A1 that persisted during active mineralization on day 17 (***Figure 4***). Remarkably, in contrast to the SOC, COL1A1 expression was not detected during active mineralization in the FH. In agreement with delayed chondrocyte maturation at the FH of *Tshr⁻/⁻* mice, the expression domain of COL2A1 was expanded, while COL10A1 was decreased compared with stage matched controls (***Figure 4***). Thus, the collagen expression profile is consistent with not only the delayed maturation at the FH but also mineralization occurring in a COL1A1-negative environment, unlike the SOCs at the distal end.

The progressive remodeling of collagens associated with distinct phases of chondrocyte maturation is principally achieved by enzymatic degradation. The key enzyme to preferentially target COL2A1 destruction is MMP13 (***Inada et al., 2004***). Immunostaining revealed that on day 10, MMP13 was largely expressed in a non-overlapping domain with COL2A1, while on days 17 and 21, MMP13 overlapped both COL2A1 and COL10A1 in chondrocytes, but minimally overlapped COL1A1 in bone, and remained expressed in FH of *Tshr⁻/⁻* mice (***Figure 5***). MMP9 degrades collagens expressed by more mature chondrocytes (***Stickens et al., 2004***), and while it was detected in bone tissue of all samples including strong expression at the SOC, it was not expressed in FH chondrocytes (***Figure 5***).

Osteoblast mineralization is affected by a pH balanced extracellular matrix, the function of which is in part regulated by carbonic anhydrase 2 (CAR2) (***Adeva-Andany et al., 2015***). While we found expression of CAR2 in osteoblasts at the tibia SOC and in bone beneath the FH, we did not detect its expression in FH chondrocytes (***Figure 5***). We also evaluated expression of non-collagenous extracellular matrix proteins involved in mineralization, IBSP, BGLAP2, DMP1, and SPP1. They were all expressed highly in the SOC and bone below the FH at all timepoints analyzed, and only BGLAP2 was found in day 10 FH. On days 17 and 21 BGLAP2 expression was lower in FH than in the bone matrix below the FH. While there was some positive signal for IBSP in the FH on days 17 and 21, the signal intensity was much less than seen in the bone beneath the FH chondrocytes. DMP1 was not detected

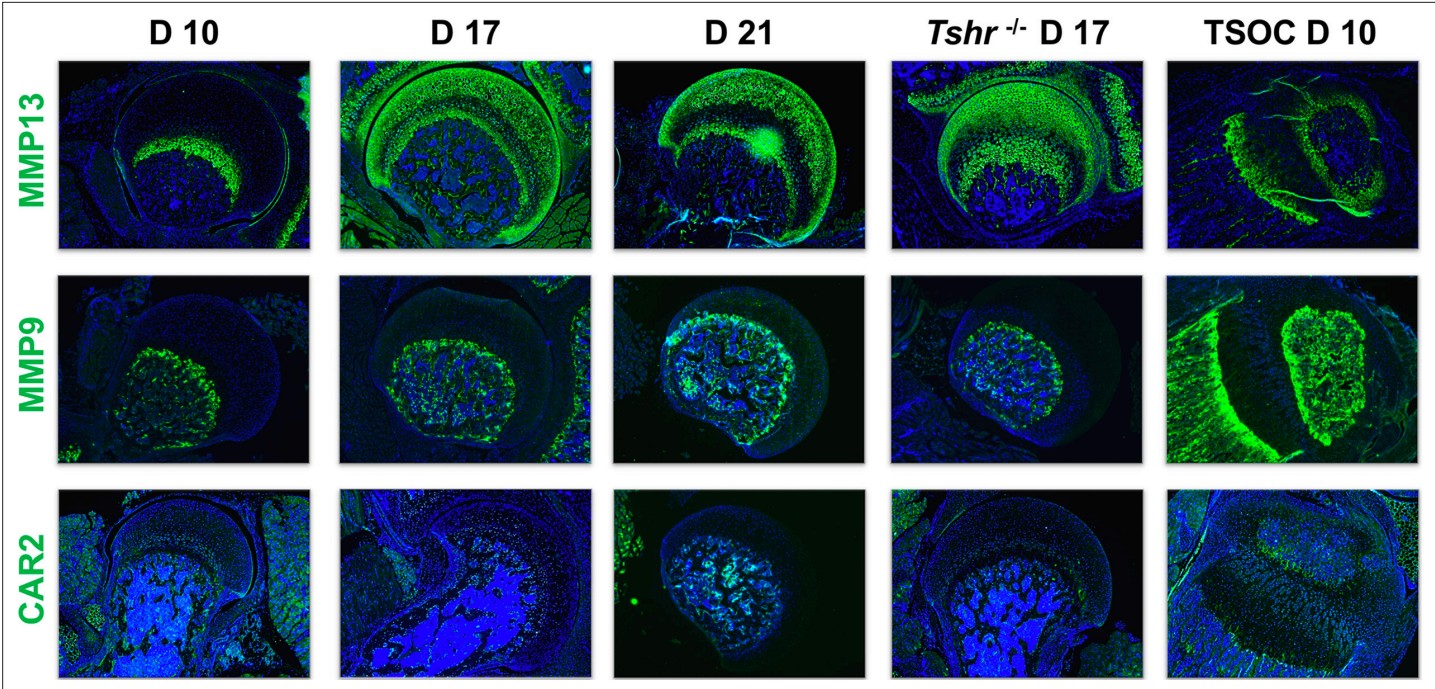

**Figure 5.** Immunohistochemical characterization of femur head (FH) development. Longitudinal sections of *Tshr*$^{+/-}$ were probed for protein expression in FH on days 10, 17, 21, proximal tibia epiphysis on day 10, and FH of *Tshr*$^{-/-}$ on day 17 by immunofluorescence for enzymes: matrix metallopeptidase MMP13, MMP9, carbonic anhydrase 2 (CAR2) stained in green; counterstained with DAPI (blue).

in the FH at any of the days evaluated, and SPP1 was expressed at high level in mineralizing chondrocytes of *Tshr*$^{+/-}$ FH on day 17 (**Figure 6**). None of these mineralization factors were expressed in *Tshr*$^{-/-}$ FH chondrocytes. ALPL activity was detected in bone and mineralizing regions of FH in *Tshr*$^{+/-}$ mice at days 17 and 21 (**Figure 6—figure supplement 1**). These results reveal that some of the players involved in mineralization are differentially expressed in the FH versus SOC.

We next determined if key transcription factors involved in chondrocyte/osteoblast differentiation (SOX9, RUNX2, SP7, DLX3, DLX5, HIF1A) are differentially expressed during mineralization of FH versus distal epiphysis. SOX9 signal was limited to chondrocytes in all regions examined. While SOX9 was not detected in hypertrophic chondrocytes in the tibia growth plate on day 10, it was present throughout the FH at all stages analyzed. RUNX2 was preferentially expressed in osteoblasts of the SOC and maturing chondrocytes at the tibia growth plate, as well as bone in the proximal femur, but not detected in FH chondrocytes until day 17 and thereafter. Both SOX9 and RUNX2 were expressed in the FH of *Tshr*$^{-/-}$ mice. Strong SP7 expression was detected at the SOC and pre-hypertrophic chondrocytes of the tibia growth plate, as well as bone of proximal tibia, but minimally expressed in FH chondrocytes until day 21, and absent in *Tshr*$^{-/-}$ FH (**Figure 7**). Both DLX3 and DLX5 were detected in mineralizing bone both proximally and distally. While DLX3 is expressed in FH chondrocytes prior to and during mineralization, DLX5 expression is weak or absent in the mineralizing region of the FH. HIF1A was expressed in the differentiating chondrocytes of both proximal femur and tibia but to a reduced extent in mineralizing bone. These results reveal that unlike the distal femur where bone is formed by endochondral ossification, SP7 likely plays a limited role in the initiation of FH chondrocyte mineralization.

Since TH is known to be critically involved in regulating chondrocyte maturation, we determined if delayed chondrocyte maturation in the FH can be explained on the basis of differences in TH receptor expression pattern in the FH versus SOC at the proximal tibia. Interestingly, we found that THRA1 was minimally expressed in proximal FH chondrocytes on day 10 but was expressed in day 10 chondrocytes and pre-hypertrophic chondrocytes of the proximal tibia (**Figure 8**). A notable gradual increase in expression was then observed on days 17 and 21, and reduced expression was seen in *Tshr*$^{-/-}$. By comparison, THRB1 was expressed on day 10 FH in subarticular chondrocytes and in an expanded domain at the FH on days 17 and 21, as well as in chondrocytes surrounding the SOC and

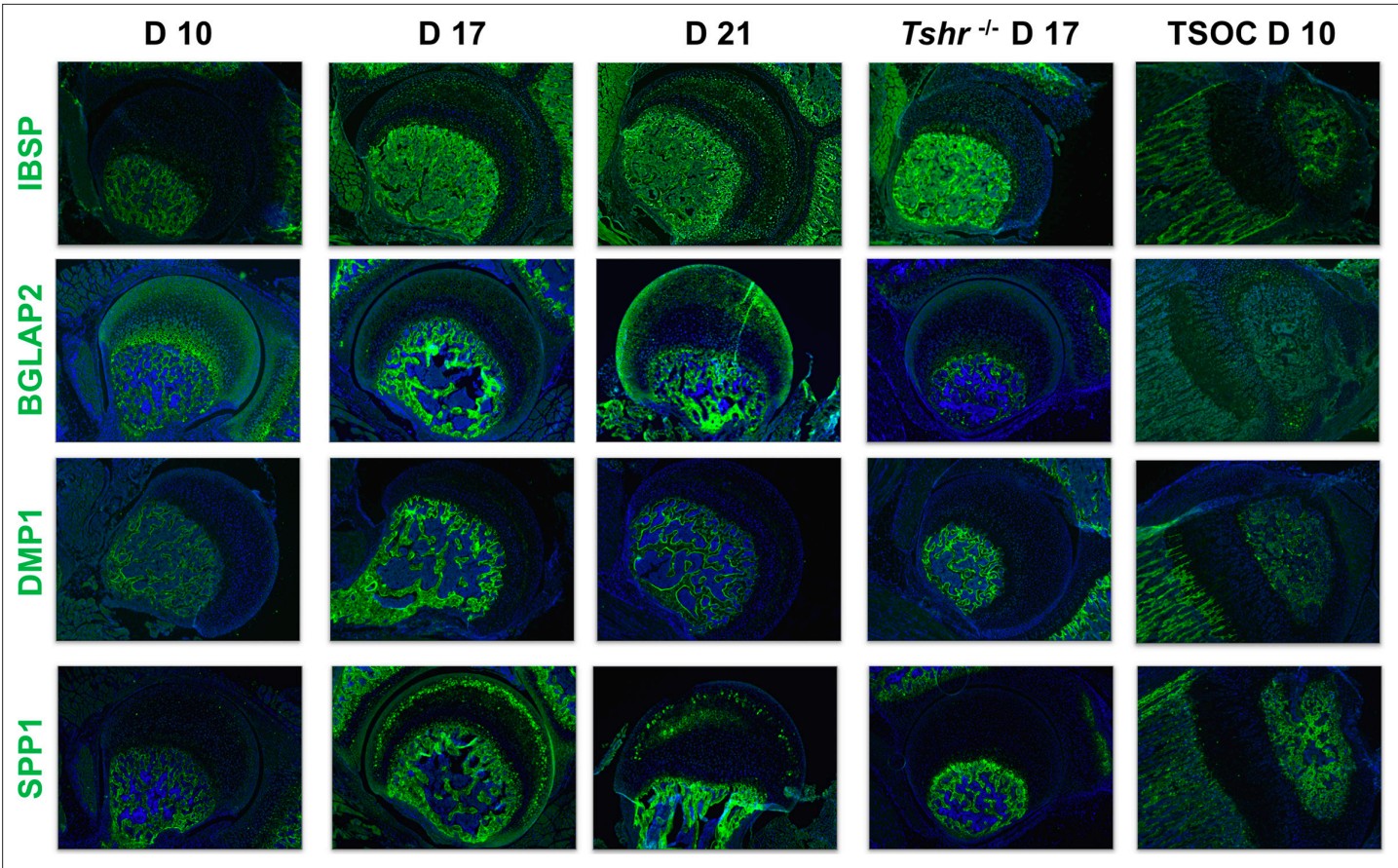

**Figure 6.** Immunohistochemical characterization of femur head (FH) development. Longitudinal sections of *Tshr*$^{+/-}$ were probed for protein expression in FH on days 10, 17, 21, proximal tibia epiphysis on day 10, and FH of *Tshr*$^{-/-}$ on day 17 by immunofluorescence for non-collagenous extracellular matrix proteins: IBSP, BGLAP2, dentin matrix protein 1 (DMP1), and SPP1 stained in green; samples were counterstained with DAPI (blue).

The online version of this article includes the following figure supplement(s) for figure 6:

**Figure supplement 1.** Alkaline phosphatase (ALPL) activity in femur head (FH) on days (D) 10, 17, 21, proximal tibia epiphysis on day 10 of *Tshr*$^{+/-}$, and FH of *Tshr*$^{-/-}$ on day 17 FH (red stain).

in pre-hypertrophic chondrocytes of the growth plate in the proximal tibia (*Figure 8*). Consistent with our identification of IHH as a direct target of TH action (*Aghajanian et al., 2017*; *Xing et al., 2016*), we found IHH expression in differentiating chondrocytes of both FH and tibia in a pattern that overlapped COL10A1 expression.

## Transcription factor regulation of chondroprogenitor differentiation

Next, we aimed to determine how TH affects chondrocyte differentiation under controlled culture conditions, and how perturbation of master regulator transcription factors (RUNX2, SP7) and their co-regulators (DLX3, DLX5) affect this process in the absence or presence of TH. We therefore knocked down each of these transcription factors in the ATDC5 chondroprogenitor cell line by lentiviral delivery of shRNAs targeting each factor and included a non-specific/random control shRNA. First, we measured the response of ATDC5 cells transduced with control shRNA to a selected panel of genes via RT-qPCR after 3 days of treatment with vehicle or TH in the absence or presence of ascorbic acid (AA), a known inducer of chondrocyte differentiation (*Altaf et al., 2006*; *Figure 9A*). Treatment with TH only resulted in significant repression of *Sox9* and *Col2a1*, as well as induction of *Col10a1*, and modulators of mineralization, *Ibsp*, *Spp1*, *Bglap2*, *Alpl*, and *Dmp1*. By contrast, addition of AA results in significant upregulation of *Runx2* and *Sp7*, an effect that was also observed in AA/TH treated cultures. Interestingly, compared with AA alone, treatment with AA/TH repressed *Dlx3* and induced *Dlx5* expression (*Figure 9A*), as well as *Col10a1*, *Spp1*, *Bglap2*, *Alpl*, and *Dmp1*. These data

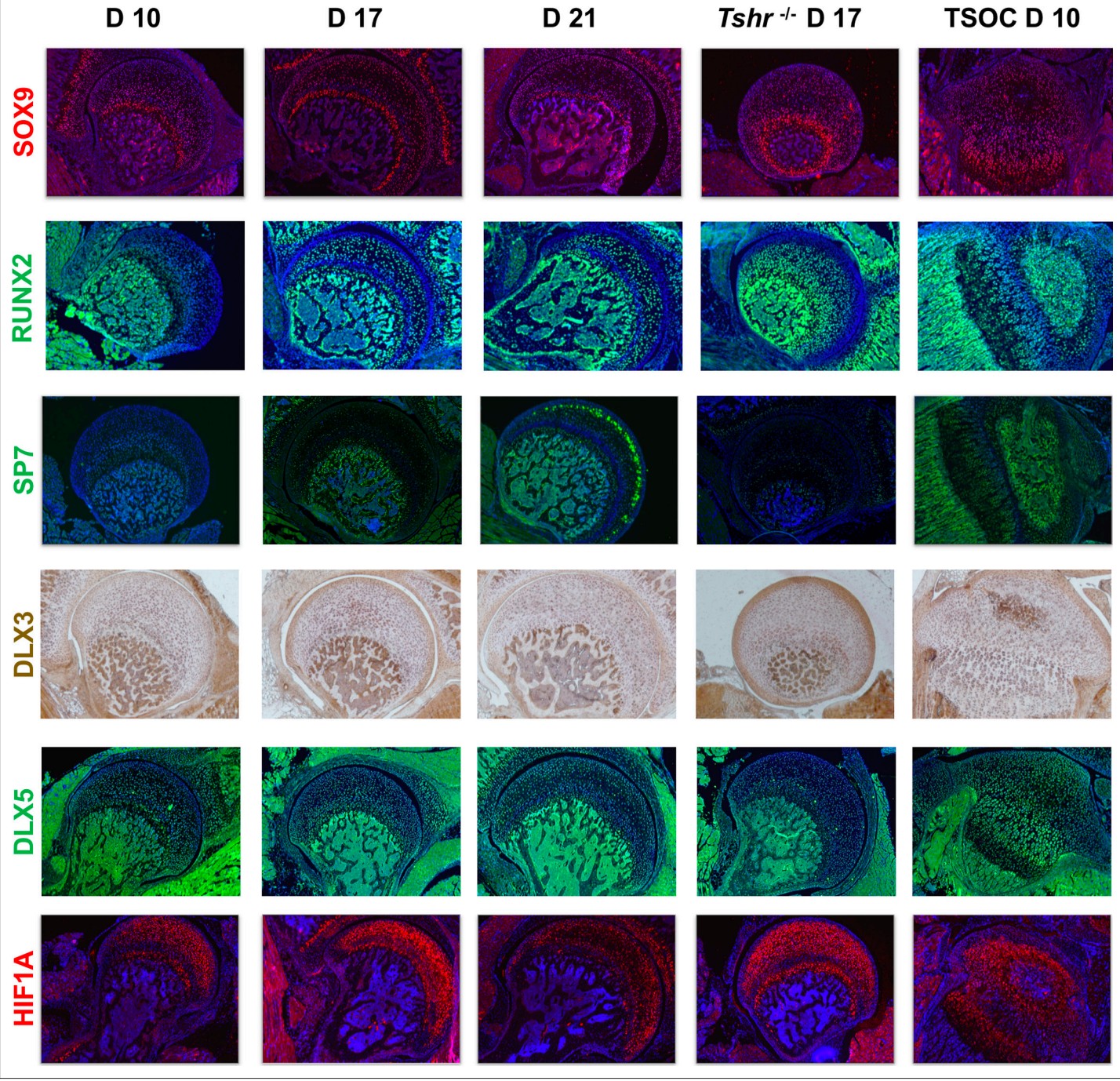

**Figure 7.** Immunohistochemical characterization of femur head (FH) development. Longitudinal sections of *Tshr*[+/-] were probed for protein expression in FH on days 10, 17, 21, proximal tibia epiphysis on day 10, and FH of *Tshr*[-/-] on day 17 by immunofluorescence for transcription factors: SOX9, HIF1A are stained in red. RUNX2, SP7, and distal-less homeobox 5 (DLX5) are stained in green. Immunofluroescence images counterstained with DAPI (blue). DLX3 stained in brown by colorimetric immunohistochemistry.

indicate that TH in general promotes expression of genes involved in chondrocyte maturation and bone mineralization in chondroprogenitors, although TH effects vary in some instances depending on whether AA is present.

The consequence of knockdown of individual transcription factors on various targets is shown in *Figure 9B–F*. The knockdown of intended targets was validated by data shown in *Figure 9B*, *Figure 9—figure supplement 1*. In the absence of TH and/or AA treatment, RUNX2, SP7, and DLX3

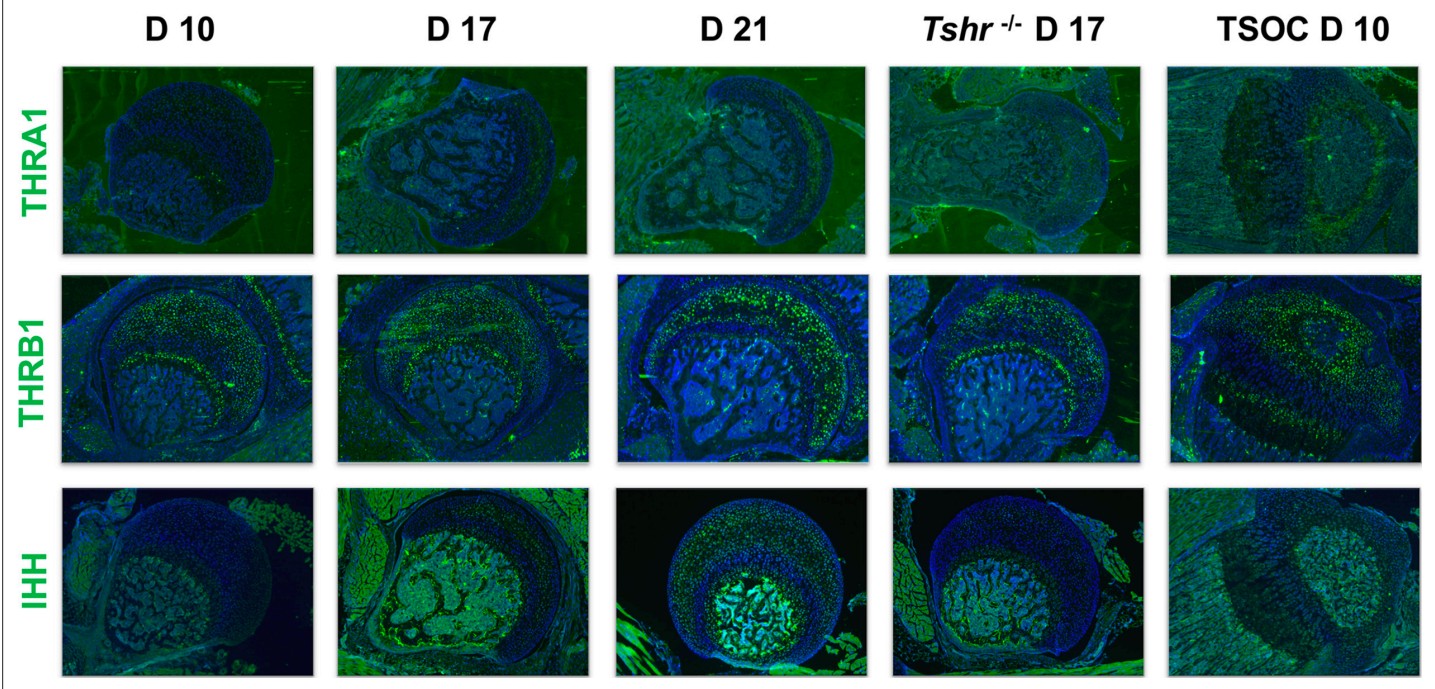

**Figure 8.** Immunohistochemical characterization of femur head (FH) development. Longitudinal sections of *Tshr*[+/-] were probed for protein expression in FH on days 10, 17, 21, proximal tibia epiphysis on day 10, and FH of *Tshr*[-/-] on day 17 by immunofluorescence for thyroid hormone response factors: thyroid hormone receptor A1 (THRA1), THRB1, IHH stained green. Counterstained with DAPI (blue).

positively regulate markers of chondrocyte maturation, *Col10a1*, *Mmp13*, but seem to inhibit expression levels of markers of matrix mineralization, *Ibsp*, *Spp1*, *Dmp1*, and *Mgp*. Interestingly, expression levels of bone formation markers, *Bglap2* and *Alpl,* are regulated in an antagonistic fashion by DLX3 (negative) and DLX5 (positive). As expected, knockdown of RUNX2 results in *Sp7* downregulation, but not vice versa (*Nakashima et al., 2002*).

Since histological data revealed that SP7, CAR2, and TRAP/ACP5 are all expressed in the distal femur but not in the FH, we evaluated whether SP7 regulates expression of these enzymes. Validation of loss and gain of function approaches was confirmed by expected effect on *Sp7* mRNA in basal conditions, and the results revealed that while SP7 perturbation positively regulates *Car2*, it does not affect *Acp5* (*Figure 9—figure supplement 2*). This suggests SP7 contributes to CAR2 expression enabling bone mineralization at the distal femur.

Next, we evaluated whether and how the four transcription factors alter TH's effect on regulation of chondrocyte differentiation by comparing changes elicited by shRNA for each transcription factor with control shRNA while in TH with and without AA (*Figure 9B-F*, *Table 1*). RUNX2 positively regulates expression of *Sox9*, *Sp7*, *Dlx3*, and *Dlx5* (*Figure 9C*), DLX3 negatively regulates expression levels of *Sp7* and *Dlx6* (*Figure 9E*). In the presence of TH, SP7 negatively regulates *Dlx3* expression (*Figure 9D*) while DLX5 positively regulates *Dlx3* expression (*Figure 9F*). While TH treatment alone represses *Col2a1* and induces *Col10a1* and *Col1a1*, in the presence of AA, TH promotes *Col2a1* and *Col10a1* expression, but not *Col1a1* (*Figure 9A*). In TH treated cultures, *Col10a1* expression is positively regulated by SP7 and DLX5, while RUNX2 is a negative regulator. All four transcription factors positively regulate *Col10a1* and *Col1a1* expression in AA/TH treated cultures. DLX3 and SP7 exert positive and negative effects respectively on *Col2a1* expression in the presence of TH. These data suggest that TH promotes maturation of chondrocytes by inhibiting *Sox9* expression and promoting *Col10a1* expression, which is likely co-regulated by all transcription factors examined.

TH treatment represses *Mmp13* expression regardless of AA involvement (*Figure 9A*). In TH treated cultures, *Mmp13* is positively regulated by RUNX2, SP7, and DLX3, while in AA/TH treated cultures, *Mmp13* is positively regulated by RUNX2 and DLX3 and negatively regulated by DLX5 (*Figure 9C and E*). *Mmp9* was repressed by TH but only in the presence of AA (*Figure 9A*). *Mmp9* is negatively regulated by RUNX2 and DLX3 in TH+ AA treated cultures as well as in control cultures (*Figure 9,*

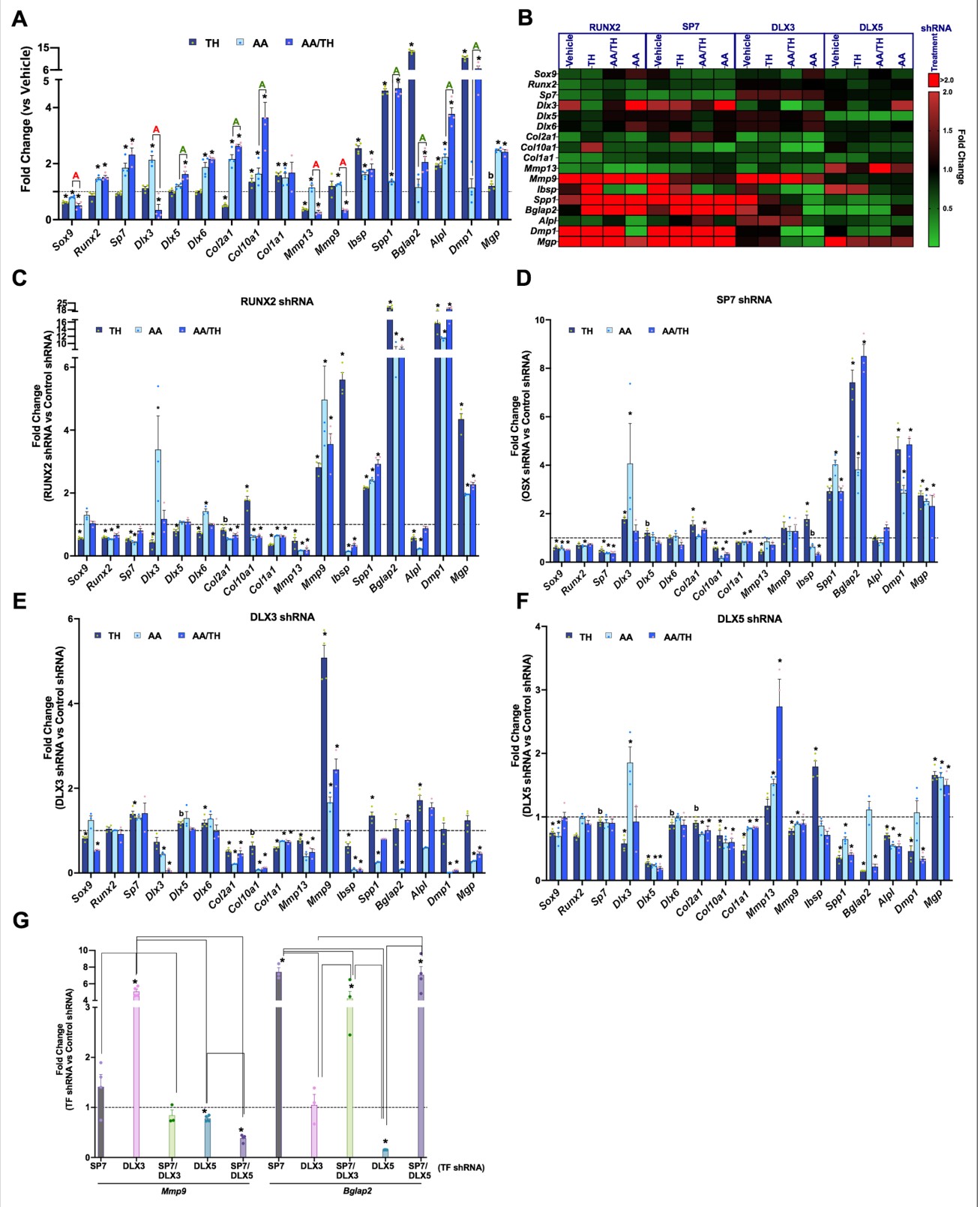

**Figure 9.** Thyroid hormone (TH) and transcription factor regulation of chondroprogenitor genes. (Panels **A–G**) Reverse transcriptase quantitative polymerase chain reaction (RT-qPCR) on day 3 for ATDC5 cell line represented as fold-change for genes labeled on X-axis. (Panel **A**) ATDC5 with control shRNA following treatment with TH, ascorbic acid (AA), or AA and TH (AA/TH). (Panel **B**) Heat map of all 17 genes analyzed in ATDC5 with shRNA for RUNX2, SP7, DLX3, DLX5. Graphed are the response to treatment with vehicle, TH, AA, or AA/TH versus control shRNA. Reduction in expression

*Figure 9 continued on next page*

*Figure 9 continued*

signifies positive regulation (green), while increased expression signifies negative regulation (red). (Panels **C–F**) Individual transcription factor knockdown versus control shRNA following treatments versus vehicle. (Panel **G**) Comparison between groups knocked down for a given transcription factor combination on X-axis versus control shRNA in TH treatment at either *Mmp9* or *Bglap2*. Statistics analyzed by t-test where * = p < 0.05 or less; b = p < 0.05–0.10, and in Panel G one-way ANOVA where comparison bars indicate p < 0.05 or less (n = 4).

The online version of this article includes the following source data and figure supplement(s) for figure 9:

**Source data 1.** Source data for *Figure 9A*.

**Source data 2.** Source data for *Figure 9C*.

**Source data 3.** Source data for *Figure 9D*.

**Source data 4.** Soure data for *Figure 9E*.

**Source data 5.** Source data for *Figure 9F*.

**Source data 6.** Source data for *Figure 9G*.

**Figure supplement 1.** Effect of individual transcription factor knockdown in vehicle treatment.

**Figure supplement 1—source data 1.** RUNX2 shRNA.

**Figure supplement 1—source data 2.** SP7 shRNA.

**Figure supplement 1—source data 3.** DLX3 shRNA.

**Figure supplement 1—source data 4.** DLX5 shRNA.

**Figure supplement 2.** Regulation of mineralization enzymes by SP7.

**Figure supplement 2—source data 1.** SP7 OE and shRNA.

**Figure supplement 3.** Thyroid hormone (TH) regulation of *Sp7* and *Dlx3* expression in chondrocytes ex vivo.

**Figure supplement 3—source data 1.** FH vs DF chondrocyte response to TH.

*Figure 9—figure supplement 1*). These results indicate that transcription factor regulation of *Mmp13* and *Mmp9* is the same in the presence or absence of TH, but RUNX2 and DLX3 impede TH-mediated repression of *Mmp13*, while DLX5 antagonizes the DLX3 effect, and further promotes TH-mediated repression of *Mmp13*. Also, RUNX2 and DLX3 further promote the negative regulation of *Mmp9* by TH observed in the presence of AA.

TH-induced expression of mineralization modulators *Ibsp*, *Spp1*, *Bglap2*, *Alpl*, and *Dmp1*, and in this condition, RUNX2 and SP7 appear to negatively regulate *Ibsp* and *Spp1* expression, in collaboration with DLX5 at *Ibsp* and DLX3 at *Spp1* (*Figure 9C-F*). It is interesting that in this condition DLX3 positively regulates *Ibsp*, while DLX5 positively regulates *Spp1*, which demonstrates their antagonism of key modulators of mineralization in the presence of TH. Similarly, in the presence of TH, *Alpl* is positively regulated by RUNX2 and DLX5 and negatively by DLX3. However, in some instances, only one of these co-regulators is active. Such is the case at *Dmp1* which is negatively regulated by RUNX2 and SP7, yet positively regulated by DLX5. DLX5 also promotes TH-mediated regulation of the *Bglap2* gene, which shows the most robust response to TH treatment in chondrocytes. By contrast, AA effect on *Bglap2* gene expression was mediated by DLX3. Overall, these data reveal a complex interplay of transcriptional circuits in ATDC5 chondrocytes treated with TH and/or AA.

Based on the known interaction between SP7 and DLX family members and the predicted DLX-mediated recruitment of SP7 to osteoblast enhancers during osteoblast specification (*Hojo et al., 2016*), we determined DLX effects in the context of whether SP7 is present or absent. *Figure 9G* shows that the knockdown effect of DLX3 or DLX5 on expression of *Mmp9* and *Bglap2* in the presence of TH is differentially affected depending on whether SP7 is present or absent. The inhibitory effect of DLX3 on *Mmp9* expression is completely lost in the absence of SP7. However, DLX5 but not DLX3 mediates TH effects on *Bglap2* expression in the presence of SP7, and the positive effect of DLX5 on *Bglap2* expression is lost when SP7 is absent. Overall, these results show that the contribution of gene regulation by co-regulators in response to TH is dependent on master regulators.

Finally, by immunofluorescence we observed higher DLX3 expression at the FH than distal femur, while the inverse relationship was observed for SP7. Since this implies critical roles for DLX3 and SP7 in differential regulation of chondrocyte fates at these two sites, we evaluated whether TH differentially regulates their expression. FH and distal femurs were isolated from control mice on day 7 and cultured in the presence of TH or vehicle and evaluated on day 22 of culture by RT-qPCR for *Sp7* and

**Table 1.** Comparison of genes regulated by different transcription factors with a p-value < 0.05 from *Figure 9* panel B after treatment with vehicle, TH, AA + TH, or AA.

'Negative' indicates knockdown of a given transcription factor results in a significant increase of gene in question, while 'positive' indicates the opposite. 2 = RUNX2; 3 = DLX3; 5 = DLX5; 7 = SP7/OSX shRNAs.

| | Vehicle treatment | | TH treatment | | AA + TH treatment | | AA treatment | |
|---|---|---|---|---|---|---|---|---|
| | Negative | Positive | Negative | Positive | Negative | Positive | Negative | Positive |
| *Sox9* | | 2,3,5 | | 2,3,5,7 | | 7 | | 5,7 |
| *Runx2* | | | | | | | | 7 |
| *Sp7* | 3 | 2 | 3 | 2 | | | | 2 |
| *Dlx3* | | 7 | 7 | 2,5 | | | 2,5,7 | |
| *Dlx5* | 3 | | | 2 | | | | |
| *Dlx6* | | | 2 | 3 | | | 2 | |
| *Col2a1* | | 2,3 | 7 | 3 | 7 | 2,3,5 | | 2,3,5 |
| *Col10a1* | | 2,3,7 | 2 | 5,7 | | 2,3,5,7 | | 2,3,5,7 |
| *Col1a1* | | 2,3,5 | | 2,3,5 | | 2,3,5,7 | | 2,3,5,7 |
| *Mmp13* | 5 | 2,3,7 | | 2,3,7 | 5 | 2,3 | 5 | 2,3 |
| *Mmp9* | 2,3 | | 2,3 | 5 | 2,3 | | 2,3 | 5 |
| *Ibsp* | 2,3 | | 2,5,7 | 3 | | 2,3,7 | | 2,3 |
| *Spp1* | 2,3,7 | 5 | 2,3,7 | 5 | 2,7 | 5 | 2,7 | 3,5 |
| *Bglap2* | 3 | 5 | 2,7 | 5 | 2,3,7 | 5 | 2,7 | 3 |
| *Alpl* | 3 | 7 | 3 | 2 | | 5 | | 2,5 |
| *Dmp1* | 2,7 | | 2,7 | 5 | 2,7 | 3,5 | 2,7 | 3 |
| *Mgp* | 2,7 | | 2,5,7 | | 2,5,7 | 3 | 2,5,7 | 3 |

*Dlx3*. Results show that TH preferentially induced *Sp7* in the distal femur and *Dlx3* in the FH, while TH negatively regulates *Dlx3* at the distal femur (*Figure 9—figure supplement 3*). This suggests FH chondrocytes may be more permissive for TH-mediated regulation of *Dlx3* than *Sp7*, while distal femur chondrocytes are more permissive to TH-mediated *Sp7* expression.

## Discussion

The salient features of our study are as follows: (1) To our knowledge, this is the first demonstration that TH provides a fundamental input for the timely formation of the proximal femur, and in particular, for chondrocyte maturation and cartilage mineralization at the FH. (2) Overall, this study provides a mechanistic framework to understand the process of cartilage mineralization and how it differs from the endochondral bone formation process. (3) Our understanding of the regulatory molecules and cellular processes involved in ossification of cartilage during normal development may provide important clues to the understanding of components involved in pathological mineralization such as that seen in vascular tissues, and, thereby, provide an opportunity to identify strategies to diagnose and correct soft tissue calcifications.

Our initial characterization of the hip joint in hypothyroid *Tshr*[-/-] mutant animals led us to investigate the chronological events involved with FH maturation and mineralization. Intriguingly, although TH levels rise systemically, distal knee epiphyseal chondrocytes respond early and undergo endochondral ossification producing an SOC, while their proximal FH counterparts experience a delay in maturation, that is exacerbated in *Tshr*[-/-] mice. Our findings that TH injections rescue timely mineralization at the FH of *Tshr*[-/-] mice provides evidence that TH provides a direct input into this process. Moreover, the detection of THRA1 and downstream genes in FH in euthyroid mice on day 17, when we observe the

earliest onset of FH mineralization, suggests that TH provides a crucial signal that initiates this process. Indeed, addition of TH to chondrogenic progenitors in culture results in the early induction of several positive and negative modulators of mineralization. Future studies are needed to identify the mechanisms for the delayed expression of THRA1 at the FH compared to distal femur and proximal tibia. It is also possible that TSH might provide auxiliary inputs that regulate chondrocytes differentially as seen in osteoblasts and osteoclasts (*Abe et al., 2003*), since TSHRs have been detected in growth plate chondrocytes (*Endo and Kobayashi, 2013*), making this question worthy of future studies.

In this study, we have identified interesting differences in the expression levels of extracellular matrix components, enzymes involved in matrix degradation and mineralization, growth factors, as well as transcription factors between FH and distal femur during the period when active mineralization occurs in these two regions. While Safranin O and toluidine blue staining revealed evidence for the presence of cartilage in the mineralized region of the FH at day 21, cartilage staining was absent in the mineralized tissue of the distal femur on day 10 (*Figure 2*). Accordingly, while COL10A1 was abundantly present in the mineralizing tissue of the FH on day 21, COL10A1 was absent at the SOC of the distal epiphysis. By contrast, COL1A1 was abundantly present in the mineralized tissue of the distal epiphysis but was totally absent in the mineralized tissue of the FH. In terms of enzymes, MMP9, CAR2, and TRAP/ACP5 were present in the mineralizing region of the distal epiphysis but absent in the FH. MMP9 was identified as a direct target of SP7 (*Yao et al., 2019*). Accordingly, while we found strong SP7 expression in the mineralizing tissue at the distal epiphysis, there was little or no signal for SP7 in the mineralizing region of the FH. Similarly, DLX5 was strongly expressed in the mineralizing tissue of the distal epiphysis but not in the FH. Also, SP7 gain and loss of function experiments in ATDC5 chondrocytes revealed that SP7 is upstream of bone matrix genes, and data from ex vivo studies demonstrate that TH regulation of SP7 in distal but not FH chondrocytes suggest that SP7 functions at the distal but not proximal end of the femur.

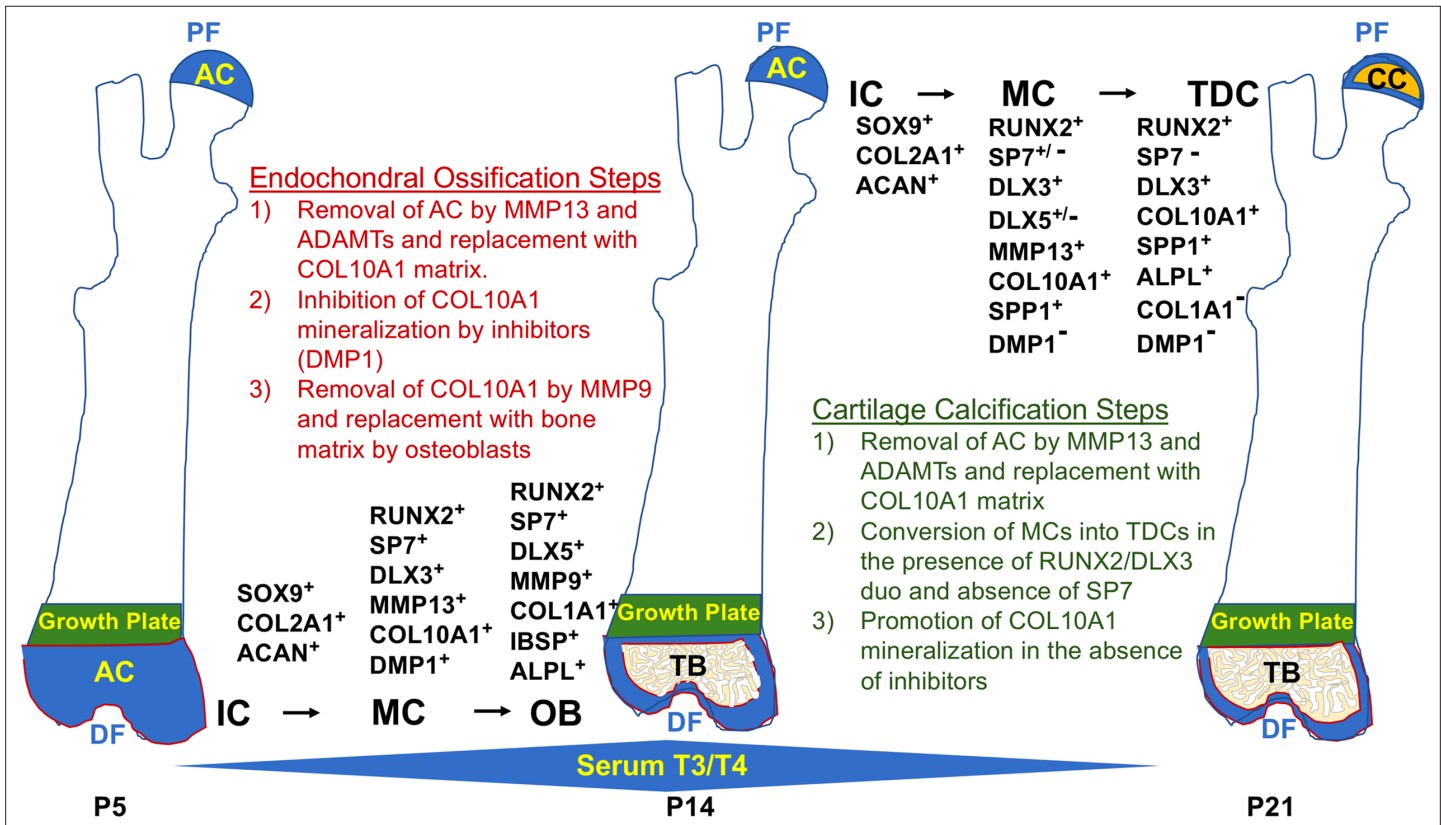

**Figure 10.** Spatiotemporal model of proximal-distal femoral chondrocyte fate acquisition. Please see text in the discussion for explanation. Abbreviations: PF, proximal femur; AC, articular cartilage; CC, calcified cartilage; DF, distal femur; IC, immature chondrocyte; MC, mature chondrocyte; TDC, terminally differentiated chondrocyte; OB, osteoblast; TB, trabecular bone; P5–21, postnatal day; T3/T4, thyroid hormone.

To understand the molecular mechanisms for TH regulation of chondrocyte maturation and bone formation, we evaluated the consequence of knockdown of master regulators of endochondral bone formation (RUNX2 and SP7) and their co-regulators (DLX3 and DLX5) on TH-induced changes in expression levels of markers of chondrocytes and osteoblasts using ATDC5 chondrocytes. We also evaluated the role of these transcription factors in mediating TH effects in the presence of AA, a known inducer of chondrocyte differentiation (*Altaf et al., 2006*; *Newton et al., 2012*). Our data as summarized in *Table 1* reveal that while RUNX2 is the main positive regulator of SP7, as expected, we identified DLX3 as a negative regulator of SP7. Interestingly, both COL10A1 and COL1A1 expression are positively regulated by all four transcription factors in AA and AA+TH treated cultures. While MMP9 expression was negatively regulated by RUNX2 and DLX3, its expression was positively regulated by DLX5. Furthermore, the negative effect of TH on MGP expression was mediated by RUNX2, DLX5, and SP7, while the positive robust effects of TH on *Bglap2* expression was mainly mediated by DLX5. Additionally, our data involving knockdown of SP7 together with DLX3 or DLX5 (*Figure 9G*) reveal that the positive effect of DLX5 on *Bglap2* expression was dependent on whether or not SP7 was present, thus revealing that interactions between SP7 and DLX factors contribute to transcriptional regulation of bone matrix genes.

Based on the findings presented in this manuscript and the known role of TH in the regulation of endochondral bone formation, we propose the following model to explain the divergent contribution of chondrocytes to endochondral ossification at the distal femur versus direct mineralization of COL10A1 matrix by chondrocytes at the proximal FH (*Figure 10*). Despite the prolonged status of an articular chondrocyte-like immature state at the FH, the initial shift toward differentiation is the same at both ends, whereby COL2A1 and ACAN are degraded by MMP13 and ADAMTs and replaced with COL10A1 by mature chondrocytes. In this mature state, COL10A1 mineralization is inhibited distally by TH-mediated induction of mineralization inhibitors such as DMP1 and MGP, allowing subsequent degradation of COL10A1 by MMP9, clearing the way for mineralization of COL1A1 secreted by osteoblasts. By contrast, at the proximal FH, an RUNX2/DLX3 duo represses MMP9 expression, protecting COL10A1 from degradation. RUNX2 also represses expression of mineralization inhibitors, DMP1 and MGP, thus allowing progression of COL10A1 mineralization. Lack of COL1A1 in the FH at the time of mineralization is consistent with the prediction that COL10A1 represents the major matrix component that is mineralized at this site. However, it remains to be seen if there are other matrix components (e.g. SPP1) that also contribute to FH mineralization. During SOC formation at the distal femur, TH induces expression of SPP1, IBSP, and BGLAP2 that are primarily mediated by DLX5. Thus, we postulate that while mature chondrocytes transdifferentiate into osteoblasts in the presence of an SP7 and DLX5 duo, which produces bone matrix that promotes endochondral bone formation at the SOC of the distal femur, RUNX2 in the absence of SP7 may interact with DLX3 to promote terminal differentiation of mature chondrocytes and subsequent cartilage mineralization at the FH.

Both master regulators of ossification, RUNX2 and SP7, are involved in chondrocyte maturation in different areas that produce bone such as POCs, secondary ossification, and growth plate. Although RUNX2 is upstream of SP7 and both may be involved in regulation of genes that promote maturation (*Artigas et al., 2014*), at the FH we find that SP7 is not notably expressed. Chondrocyte-specific knockout of SP7 has been reported to result in expanded hypertrophic chondrocyte mineralization (*Jing et al., 2014*; *Zhou et al., 2010*), demonstrating that SP7 is not required for chondrocyte mineralization, and its timely absence may even affect the fate of RUNX2+ hypertrophic chondrocytes. Incidentally, forced expression of RUNX2 is sufficient to increase the rate of chondrocyte maturation and induce ectopic chondrocyte mineralization in vivo (*Takeda et al., 2001*). Also, SP7 has been shown to be a positive regulator of COL1A1 and its interaction with DLX5 was shown to be critical for osteoblast specification (*Hojo et al., 2016*; *Ortuño et al., 2013*). Our findings support the model that an SP7/DLX5 duo contributes to transdifferentiation of chondrocytes to osteoblasts at the distal femur during endochondral bone formation, and the limited activity of SP7 at the proximal femur may partly explain the difference in fates there.

We sought to understand why chondrocyte maturation at the proximal FH is delayed compared with the distal femur. One possibility is a difference in endothelial vasculature, but distal femur chondrocytes mature in response to TH prior to expression of VEGFA (*Aghajanian et al., 2017*), limiting this option. We searched whether VEGFA is expressed in the FH but did not detect it (data not shown), consistent with results reported by *Cole et al., 2013*. Therefore, the presence

of HIF1A+/VEGFA- chondrocytes support the notion that HIF1A does not regulate VEGFA expression in this context, but is likely promoting collagen hydroxylation or contributing to chondrocyte metabolism (*Bentovim et al., 2012*). Another possibility is that TH might regulate growth factor expression differentially in chondrocytes at the two ends of the femur, as shown by our data on *Tgfa* (*Figure 3A*). It is equally possible that TH-mediated activation or repression of signals from adjacent structures can affect the timing of maturation and mineralization at the proximal FH. Alternatively, TH activity might be subdued early at the FH. Indeed, we found that the TH receptor, THRA1, was highly expressed distally on day 10, but minimally at the FH in the same mice at postnatal day 10. In this regard, a complementary pattern was noted for the expression of IHH, a direct target of TH at both ends, supporting the likelihood that delayed chondrocyte development at the FH is caused by reduced TH activity. Addressing this question further will be worthwhile for future investigations.

In this study we exclusively analyzed the early stages of FH development in mice, and while there are differences between mouse and human FH formation (*Cole et al., 2013*), there are also crucial similarities. For instance, a time delay in the development of proximal HF compared to distal femur has been demonstrated in humans as in the case of mice (*Windschall et al., 2016*). Furthermore, hypothyroidism has been proposed as one of the major causes of slipped capital femoral epiphysis, revealing an important role for TH in the timely formation of FH in humans (*Kadowaki et al., 2017*; *Wells et al., 1993*).

The potential clinical relevance of our findings are as follows: it is known that physiological mineralization is necessary for the formation of skeletal tissues and is restricted to specific sites in skeletal tissues including cartilage, bone, and teeth. Mineralization can also occur in an uncontrolled or pathological manner in many soft tissues including cardiovascular, kidney, and articular cartilage leading to morbidity and mortality. Recent studies focused on the underlying mechanisms for vascular calcification have shown that components regulating physiological mineralization are also present in areas of pathological mineralization (*Bourne et al., 2021*; *Tesfamariam, 2019*), suggesting that mechanisms for pathological mineralization may be a recapitulation of what happens during normal development. Therefore, studies focused on the understanding of regulatory molecules and cellular processes involved in ossification of cartilage and bone tissues during normal development may provide important clues toward the understanding of components involved in pathological mineralization. Our further confirmation of the role of molecular signals and mechanisms that contribute to TH effects on cartilage versus bone mineralization at FH and distal femur, respectively, could lead to the development of novel strategies for prevention and treatment of osteoarthritis and other soft tissue calcification disorders.

## Materials and methods
### Mouse model

We obtained the *Tshr^hyt^* mouse strain from Jackson Laboratories (Bar Harbor, ME). Animals were inbred and tail snip extracted DNA was genotyped for *Tshr* mutation by RT-qPCR. Mice were housed in standard housing conditions at the VA Loma Linda Healthcare System Veterinary Medical Unit (Loma Linda, CA). All procedures were approved by the Institutional Animal Care and Use Committees of the VA Loma Linda Healthcare System (Permit #0029/204). At time of sacrifice, mice were anesthetized in isoflurane, then exposed to $CO_2$ prior to cervical dislocation, and bones dissected for further processing. For TH replacement experiments, genotyped mice were injected intraperitoneally with 1 µg T3 and 10 µg thyroxine (T4) (Sigma-Aldrich), or an equal volume of vehicle (5 mM NaOH) for 10 days (on days 5–14). Bones of mice studied on day 10 were injected on days 5–9 and sacrificed on day 10. *Tshr* mice do not show gender differences until 5 or 6 weeks. Therefore, mice were pooled regardless of gender in all analysis.

### X-ray

Femur X-rays of anesthetized hypothyroid *Tshr^-/-^* and euthyroid *Tshr^+/-^* were obtained from Faxitron Radiography system (Hologic, Bedford, MA) at postnatal day 21 using 20 kV X-ray energy for 10 s.

## Micro-CT

Proximal femurs were evaluated by µCT (viva CT40; Scanco Medical, Switzerland) as described previously (*Xing et al., 2014*). Proximal femurs were isolated from postnatal day 21, fixed in 10% formalin overnight (ON), then washed and imaged in phosphate buffered saline (pH 7.4). Bones were scanned by X-ray at 55 kVp volts at a resolution of 10.5 µM/slice. Images were reconstructed using the 2D and 3D image software provided by Scanco Viva-CT 40 instrument (Scanco USA, Wayne, PA).

## Nano-CT

Proximal and distal femurs of *Tshr$^{+/-}$* were scanned at postnatal days 10, 14, 17, and 21 using a nano-CT at a voxel dimension of 0.3 µm (VersaXRM-500; Xradia, Pleasanton, CA). Images were captured using software provided by Xradia.

## Histology

Mouse femurs were fixed in 10% formalin overnight, washed, decalcified in 10% EDTA (pH 7.4) at 4°C for 7 days with shaking and embedded in paraffin for sectioning. Longitudinal sections of the proximal and distal femur were stained with various stains using standard procedures.

## Immunohistology

Longitudinal 5 µM sections at regions of interest shown in figures were obtained by immunofluorescence following standard protocols. Both paraffin and cryosections were processed. Dissected bones were fixed for 3 days in either 10% formalin (paraffin) or 4% paraformaldehyde (cryosections) at 4°C, followed by 1 week of de-calcification in 20% EDTA in PBS buffer (pH 7.5). Cryosectioned samples were embedded and sectioned in OCT (FisherScientific, 23-730-571). Paraffin sections were deparaffinized in Histochoice clearing reagent (Amresco, H103-4L), gradually re-hydrated in ethanol through PBS, then permeabilized in 0.5% Triton X-100 (SIGMA, T-9284) for 15 min at room temperature (RT), rinsed in PBS, followed by an antibody-specific antigen retrieval approach (see *Supplementary file 1*). Cryosections were processed identically except they were thawed to RT then started at the permeabilization step. Following antigen retrieval, tissue sections were blocked in 2.5% serum and incubated in primary antibodies ON at RT. Commercial species-specific secondary antibodies were used (VECTOR labs, DI-1788 or DI-3088), and sections were counterstained with DAPI (Invitrogen, D1306). Colorimetric immunohistochemistry followed same steps as for immunofluorescence except (1) endogenous peroxidase was quenched with 3% H$_2$O$_2$ prior to permeabilization (2) biotinylated goat anti-rabbit HRP secondary antibody was added at 1:200 (Vector BA-1000), followed by a 1:200 dilution of streptavidin-HRP (Vector: SA-5004), and detected by enzyme reaction with Betazoid DAB chromogen (BIOCARE BDB2004H). The Vector lab's mouse on mouse (MOM) kit (BMK-2202) was used with mouse primary antibodies according to manufacturer's instructions. Other VECTOR reagents used with MOM kit: Avidin/Biotin block, #SP-2001; Fluorescein Avidin #DCS A-2011.

## Microscopy

Immunofluorescence images were obtained on a 5× dry objective on a Leica Digital Microscope DMI6000B with Leica Application Suite X software. Histological and colorimetric immunohistochemical images were obtained on an Olympus microscope with an Olympus DP72 camera with DP2-BSW software. All immunohistological results were processed together with consecutive sections that either received no primary antibody or species-specific IgG antibody. These were imaged at identical parameters as sections probed with antibodies.

## ALPL histochemistry

ALPL activity assay was performed on cryosectioned samples as described (*Miao and Scutt, 2002*). Sections were warmed to RT, OCT compound washed out, then incubated in ALPL buffer (6.055 g Tris; 5.84 g NaCl; 0.147 g CaCl$_2$•2 H$_2$O; 0.372 g KCl; 0.203 g MgCl$_2$•6 H$_2$O in 1 L H$_2$O pH 8.6) containing 1% magnesium chloride at 4°C O/N. Next day, samples were directly transferred to ALPL buffer + substrate (0.2 mg/mL naphthol AS-MX phosphate [Sigma-Aldrich, N6125-1G] and 0.4 mg/mL Fast Red violet LB [Sigma-Aldrich, F3381-1G]). Reaction was monitored, sections were rinsed with PBS, and imaged immediately. All samples received identical reaction time.

## Cell Culture

The ATDC5 mouse chondrocyte cell line was purchased from the American Type Culture Collection (Manassas, VA), tested negative for mycoplasm, and validated by upregulation of chondrocyte differentiation markers in response to chondrocyte differentiation protocols. Cells were maintained in DMEM-F12 medium containing 5% FBS, penicillin (100 U/mL), and streptomycin (100 µg/mL) at 5% $CO_2$ in normoxic conditions at 37°C. Cells were incubated in the presence of serum-free DMEM-F12 medium containing 0.1% bovine serum albumin and antibiotics for 24 hr prior to treatment with 10 ng/mL T3 (Sigma-Aldrich), 50 µg/mL AA and/or 10 mM β-glycerol phosphate (BG). Vehicle control indicate cells treated with BG only. shRNA knockdown was achieved by transduction of Mission Lentiviral particles (Millipore, Sigma): control shRNA- Cat#SHC002V; RUNX2 NM_009820, Cat#TRCN0000095590; DLX3 NM_010055, Cat#TRCN0000430532; DLX5 NM_010056.2 Cat#TRCN0000428940; Sp7 NM_130458 Cat#TRCN0000423959. SP7 overexpression studies were performed as described (*Lindsey et al., 2019*).

## Ex vivo chondrocyte culture

Proximal HF and distal femur epiphyses were surgically isolated from 7-day-old Sp7 floxed homozygous mice and cells isolated via enzymatic digestion with collagenase 1 (2 mg/mL) and hyaluronidase (1 mg/mL) or collagenase D (2 mg/mL) as described (*Lindsey et al., 2019*). Femoral head and epiphyseal chondrocytes infected with GFP adenovirus were grown in alpha MEM containing 10% fetal bovine serum, penicillin (100 units/mL) and streptomycin (100 µg/mL) and treated with TH (10 ng/mL) or vehicle for 22 days prior to RNA extraction and real-time RT-PCR.

## Real-time quantitative PCR

RNA was extracted from epiphyseal chondrocytes or ATDC5 cells in TRI reagent (Molecular Research Center INC, TR118) according to manufacturer's instructions, and purified on silica columns with E.Z.N.A. Total RNA Kit I (Omega BIO-TEK, R6834-02). Total RNA was reverse transcribed to cDNA with oligo(dT)$_{12-18}$ and Superscript IV Reverse transcriptase (Invitrogen, 18091050). A final concentration of 0.133 ng/µL was used per real-time PCR reaction with InVitrogen SYBR green (ThermoFisher, 4309155) and processed on a ViiA 7 RT-PCR system. All reactions were standardized with peptidyl prolyl isomerase A primers. Primer sequences used for RT-qPCR are listed in *Supplementary file 2*. Fold changes were calculated by the Delta Ct method, and statistics analyzed by t-test (processed on Microsoft Excel 365) or one-way ANOVA (processed on GraphPad Prism9). Error bars in all graphs indicate ± standard error of mean (SEM).

# Acknowledgements

This work was supported by National Institutes of Health Grant R01 AR048139 (SM), R21 AG062866 (SM), and Veterans Administration BLR&D Grant BX005263 (SM). SM is a recipient of Senior Research Career Scientist Award from Veterans Administration. The authors thank Veterans Administration for providing facilities to perform the work. Expert technical assistance by Catrina Godwin, Fern Baedyananda, Jasmine Lau, Subhashri Das, Heather Watt, and Nancy Lowen are greatly acknowledged.

# Additional information

#### Competing interests

Patrick Aghajanian: Patrick Aghajanian is affiliated with Fulgent Genetics The author has no financial interests to declare. Subburaman Mohan: Reviewing editor, eLife. The other authors declare that no competing interests exist.

## Funding

| Funder | Grant reference number | Author |
|---|---|---|
| National Institute of Arthritis and Musculoskeletal and Skin Diseases | AR048139 | Subburaman Mohan |
| National Institute on Aging | R21AG062866-02 | Subburaman Mohan |
| Veterans Administration BLR&D | BX005263 | Subburaman Mohan |

The funders had no role in study design, data collection and interpretation, or the decision to submit the work for publication.

## Author contributions

Gustavo A Gomez, Data curation, Formal analysis, Investigation, Methodology, Visualization, Writing – original draft, Writing – review and editing; Patrick Aghajanian, Destiney Larkin, Methodology; Sheila Pourteymoor, Data curation, Formal analysis, Methodology; Subburaman Mohan, Conceptualization, Funding acquisition, Investigation, Project administration, Resources, Supervision, Visualization, Writing – original draft, Writing – review and editing

## Author ORCIDs

Gustavo A Gomez http://orcid.org/0000-0001-9294-4276
Subburaman Mohan http://orcid.org/0000-0003-0063-986X

## Ethics

All procedures were approved by the Institutional Animal Care and Use Committees of the VA Loma Linda Healthcare System (Permit Number: 0029/204). Every effort was made to minimize animal suffering.

## Decision letter and Author response

Decision letter https://doi.org/10.7554/eLife.76730.sa1
Author response https://doi.org/10.7554/eLife.76730.sa2

# Additional files

## Supplementary files

• Transparent reporting form

• Supplementary file 1. Antibodies.used. Antigen retrieval with: HD = 2 mg/mL hyaluronidase (Sigma-Aldrich) in PBS at 37°C for 45 min. S = 10 mM sodium citrate with 2 mM EDTA and 0.05% Tween-20 (pH 6.0) at 95°C for 6 min. L = 10 mM sodium citrate with 2 mM EDTA and 0.05% Tween-20 (pH 6.0) at 60°C for 72 hr.

• Supplementary file 2. Real Time quantitative PCR primers used.

## Data availability

The numeral data used to generate figures were uploaded separately as: Figure 3 -Source Data 1-3; Figure 9 - Source Data 1-6; Figure 9- figure supplement 1- source data 1-4; Figure 9- figure supplement 2-source data 1; Figure 9- figure supplement 3- source data 1.

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
