## [Editor Report]

This paper describes the differential effects of thyroid hormone on chondrocyte maturation and mineralization at the femoral head in mice compared with the distal femur. The comprehensive set of studies were carried out in *Tshr*^*-/-*^ hypothyroid mice through radiologic and histologic methods as well as transcriptional profiling. The conclusions are of value in understanding bone growth defects during hypothyroid states.

---

## [Decision Letter]

**Decision letter after peer review:**

Thank you for submitting your article "Differences in pathways contributing to thyroid hormone effects on postnatal cartilage calcification versus secondary ossification center development" for consideration by *eLife*. Your article has been reviewed by 2 peer reviewers, and the evaluation has been overseen by me. The following individuals involved in review of your submission have agreed to reveal their identity: Jameel Iqbal (Reviewer #2) and Rauf Latif (Reviewer #2).

After consultative discussions, I have drafted this to help you prepare a revised submission.

Essential revisions:

1) The authors show that lack of expression of carbonic anhydrase and TRAP represent key differences between mineralizing cartilage versus bone. Since osterix seems to be abundantly expressed in mineralizing bone but not in cartilage, an obvious question is whether osterix expression in epiphyseal chondrocytes contributes to the expression of these enzymes at the secondary ossification centers. It will, therefore, be interesting to evaluate if carbonic anhydrase and TRAP are direct targets of osterix.

2) The authors proposed model predicts a key role for Dlx3 in cartilage mineralization but not endochondral bone ossification. It will be interesting to test if thyroid hormone differentially affects Dlx3 expression in chondrocytes-derived from the proximal femoral head versus distal femur. Such a finding would provide further support for this model.

3) There is abundant evidence in the literature that TSH has direct actions in bone cells, both osteoblasts and osteoclasts (eg Abe et al., 2003, Cell). Since TSHR haploinsufficiency is being compared with complete absence of the TSHR, the extent to which TSHR signaling might confound these actions is unclear. The authors need to discuss this in some detail as a potential caveat. Are there TSHRs on normal chondrocytes?*Reviewer #1:*

While the role of growth factors and transcription factors that contribute to conversion of cartilage to bone during endochondral ossification is well studied, relatively little is known on the cellular and molecular processes that contribute to direct calcification of cartilage. By using thyroid hormone-deficient Tshrhyt mouse model, the authors demonstrate that thyroid hormone provides a fundamental input for the timely formation of the proximal femur, and, in particular, for chondrocyte maturation and cartilage mineralization at the femoral head.

Based on the data from gene/protein expression analyses and knockdown experiments, the authors have identified a mechanistic framework for chondrocyte maturation and cartilage mineralization at the femoral head. These data provide a major conceptual advancement of our understanding of differences between cartilage and bone mineralization.

Overall, this is a well designed and conducted study. The manuscript is clearly written, and the methods and results are well described. The authors conclusions are well supported by experimental data. The authors proposed model provides novel insights into the regulatory and cellular mechanisms that contribute to differences in cartilage mineralization versus endochondral ossification.*Reviewer #2:*

Gustavo A. Gomez et al., in this paper have used the TSHRhyt mouse model to investigate whether thyroid hormone (TH) plays a key role in proximal femur head ossification. Using, X- ray, micro CT, histology, immunofluorescence and transcriptional profiling the authors arrive at the conclusion that TH controls the ossification of FH through chondrocyte maturation and mineralization. The stark differences in femur head formation in these two group of mice and restoration of mineralization of femur head (FH) in TSHR -/- mice after TH treatment compared to the distal SOC launched the authors into series of experiments to examine the pathways differences which leads to different mechanisms of proximal FH calcification and endochondral ossification. The delay in FH cartilage mineralization was followed in hypothyroid mice by histochemical staining with specific dyes and expression of genes important in mineralization. By performing impressive analysis of transcription factors and their in-vitro studies on ATDC5 cells following siRNA inhibition experiments the authors narrow down the fact that TH in general promotes expression of genes in chondrocyte maturation and bone mineralization in chondroprogenitors. By this careful and extensive evaluation the authors shown the complex molecular pathway that leads to difference in the FH chondrocyte maturation and subsequently ossification.

---

## [Author Response]

Essential revisions:1) The authors show that lack of expression of carbonic anhydrase and TRAP represent key differences between mineralizing cartilage versus bone. Since osterix seems to be abundantly expressed in mineralizing bone but not in cartilage, an obvious question is whether osterix expression in epiphyseal chondrocytes contributes to the expression of these enzymes at the secondary ossification centers. It will, therefore, be interesting to evaluate if carbonic anhydrase and TRAP are direct targets of osterix.

We have performed additional real time PCR measurements for carbonic anhydrase (CA) and *Trap* genes using RNA collected from chondrocyte cultures that overexpress or knockdown Osterix and included this data in Figure 9 Supplement 2. Our data show that while Osterix knockdown significantly reduced CA expression, its overexpression increased CA expression, thus suggesting that Osterix is a positive regulator of CA expression in chondrocytes. While *Trap* expression was reduced or increased, respectively by knockdown and overexpression of Osterix, these changes were not statistically significant.

2) The authors proposed model predicts a key role for Dlx3 in cartilage mineralization but not endochondral bone ossification. It will be interesting to test if thyroid hormone differentially affects Dlx3 expression in chondrocytes-derived from the proximal femoral head versus distal femur. Such a finding would provide further support for this model.

Our additional data as shown in Figure 9 Supplement 3 show that the expression of *Osterix* and *Dlx3* are differentially regulated by thyroid hormone treatment in chondrocytes derived from femur head and distal femur epiphysis, a finding consistent with our proposed model of Osterix and Dlx3’s role in bone and cartilage calcification.

3) There is abundant evidence in the literature that TSH has direct actions in bone cells, both osteoblasts and osteoclasts (eg Abe et al., 2003, Cell). Since TSHR haploinsufficiency is being compared with complete absence of the TSHR, the extent to which TSHR signaling might confound these actions is unclear. The authors need to discuss this in some detail as a potential caveat. Are there TSHRs on normal chondrocytes?

We thank the reviewer for pointing out the potential role of TSH in bone cells. We agree with the possibility of a direct role for TSH in endochondral ossification and/or cartilage calcification. We have cited evidence for the presence of TSH receptor in chondrocytes and addressed the potential caveat about the need for further studies to address the confounding actions of TSHR signaling in mediating the actions of thyroid hormone axis in cartilage and bone calcification processes.